# The Truman Show for Human Helminthic Parasites: A Review of Recent Advances in In Vitro Cultivation Platforms

**DOI:** 10.3390/microorganisms11071708

**Published:** 2023-06-29

**Authors:** Smita Sutrave, Martin Heinrich Richter

**Affiliations:** Department of Biological Safety, German Federal Institute for Risk Assessment (BfR), Max-Dohrn-Str. 8–10, 10589 Berlin, Germany

**Keywords:** helminths, neglected tropical diseases (NTDs), zoonosis, in vitro culture, 3D model, climate change

## Abstract

Throughout history, parasites and parasitic diseases have been humankind’s constant companions, as evidenced by the findings of tapeworm eggs in ancient, mummified remains. Helminths are responsible for causing severe, long-term, and debilitating infectious diseases worldwide, especially affecting economically challenged nations due to prevailing deficits in access to sanitation, proper hygiene practices, and healthcare infrastructure. Socio-ecological drivers, such as poverty, migration, and climate change, continue to contribute to parasites and their disease vectors being spread beyond known endemic zones. The study of parasitic diseases has had a fair amount of success leading to the development of new chemotherapeutic agents and the implementation of parasite eradication programs. However, further progress in this direction has been hampered by the challenges of culturing some of these parasites in in vitro systems for efficient availability, basic life cycle, infection studies, and effectiveness of novel treatment strategies. The complexity of the existing models varies widely, depending on the parasite and its life cycle, ranging from basic culture methods to advanced 3D systems. This review aims to highlight the research conducted so far in culturing and maintaining parasites in an in vitro setting, thereby contributing to a better understanding of pathogenicity and generating new insights into their lifecycles in the hopes of leading to effective treatments and prevention strategies. This work is the first comprehensive outline of existing in vitro models for highly transmissible helminth diseases causing severe morbidity and mortality in humans globally.

## 1. Introduction

Parasitic infections in humans are highly communicable, with disease transmission often occurring alimentarily, through contact with a contaminated environment or via insect vectors. Person-to-person transmission, when possible, occurs mainly through the oral–fecal route. Some of these infections may become chronic, lead to severe disease profiles, and can be fatal. Areas with poor living conditions, lack of hygiene, limited access to medical care, lack of infrastructure, and regions affected by natural disasters, as well as areas with civil unrest, show a higher burden of disease (BoD) associated with parasitic diseases [1,2,3,4,5,6,7,8]. Helminthiasis caused by intestinal parasitic worms is common in economically weaker parts of the world with limited access to proper sanitation. Helminth infections are characterized by the shedding of infective eggs in the stool, leading to further spread of the parasite via smear infections. However, helminths, such as *Trichinella* spp., are endemic throughout the world, with recent reports of game animals serving as reservoirs in developed nations as well [9].

Efforts have been underway to improve the disease burden caused by parasites with support from governmental and non-governmental organizations (NGOs) [10,11,12] aiming to eradicate some of these diseases through funding of research, including the development of in vitro models to better understand parasite development and, ultimately, identify strategies for treatment directly affecting the parasite.

In order to make progress toward the treatment and prevention of human parasitic diseases, there is a need for model systems that closely simulate the parasitic life cycle within the human host. Although this can generally be achieved using in vivo models, doing so at least requires mammalian hosts and can sometimes only be achieved in large mammals in order to obtain the desired parasitic life-cycle stages. Additionally, long-term pathogen maintenance in animal models is often difficult and has ethical implications. In vitro models are helpful in filling in the gaps of knowledge in parasite lifecycles, as well as pathogenicity, and provide platforms for testing anthelmintic agents while avoiding the ethical issues of in vivo studies. In the future, in vitro models may also contribute to the ready availability of desired parasite stages in relevant amounts. However, despite recent advances in in vitro culture models and technologies, developing suitable helminth in vitro cultures remains challenging. Helminths, including those that can cause disease in humans, undergo developmental phases that are not exclusively conducted in one physiological compartment of the host, but the parasite has to pass several of those compartments, e.g., travel from the gut to the musculature, to successfully propagate. Moreover, several helminthic parasites undergo stage development in different hosts. Therefore, when designing an in vitro model, it is pertinent to specifically design the in vitro environment or environments needed to achieve a certain developmental stage. Downstream of the development of an in vitro model remains the difficult task to show that the model indeed replicates human infection conditions.

Table 1 provides a measure of the disease burden due to parasitic helminths worldwide, including the number of cases, deaths, and treatments available for each parasite included in this review. However, as this review discusses the in vitro cultivation, only parasitic diseases for which an in vitro system has been described have been listed in the table. The DALY (Disability-Adjusted Life Year) metric, adopted by WHO Global Health Estimates (GHE) in order to comprehensively measure the global burden of disease (GBD) on society due to a variety of ailments starting in 2000 with the latest data generated until 2016 [13], is included in Table 1. For a given disease, one DALY is equivalent to one lost year of “healthy” life and is obtained by summing up the Years of Life Lost (YLL) due to premature death and the Years Lived with Disability (YLD) due to sub-optimal health caused by a specific disease [14].

Fundamentally, the improvement of basic hygiene is directly related to improvements in quality of life [15]. Therefore, in addition to research efforts, provision of acceptable living conditions is paramount to parasitic disease prevention and, ultimately, disease eradication among all age groups. The study of parasites in human health and disease continues to remain a major topic of interest in the public health domain and should, therefore, be given due attention. Hence, this review aims to summarize existing in vitro models for helminths known to cause severe disease in humans.

## 2. Materials and Methods

Searches of PubMed using the search terms “in vitro model” (59 hits), “in vitro culture (1489 hits), and “in vitro cultivation” (184 hits), along with the scientific name for each parasite, were used to identify the references cited in this review. The historical background literature from a timeframe spanning the years 1967–2022 was used for this study. For this review, all relevant peer-reviewed international journal article hits without limiting publication date or language were considered. Model selection was based on two criteria. The first criterion was whether the desired developmental stage could be maintained in culture for a significant amount of time, and the second criterion was whether the desired developmental stage in culture could be considered complete. Time allocation was determined with respect to the physiological survival time of the respective stage of each parasite under natural conditions.

## 3. Results

### 3.1. Helminths

Results of recently developed in vitro models for Cestodes, Trematodes, and Nematodes will be presented, respectively.

#### 3.1.1. Cestoda

The class cestoda includes tapeworms belonging to the Phylum Plathelminths. Cestodes are hermaphroditic, segmented organisms where each adult tapeworm contains both male and female sexual organs capable of self-fertilization. In the host, mature segments containing infective eggs are passed out in the stool. The lifecycles of taeniid worms involve eggs containing larval oncospheres, intermediate postoncospheral forms, and infective stages containing invaginated scolex or scolices (sometimes protoscolices) able to develop into an adult tapeworm.

##### *Taenia* *solium*

*Taenia* (*T.*) *solium*, also known as the pork tapeworm, has a worldwide distribution with infections being caused by eating insufficiently cooked pork products [16]. *T. solium* is known to cause two disease profiles, depending on the infection route. Intestinal taeniasis is a mild intestinal disease caused by the ingestion of cysts via the consumption of undercooked or raw pork products. The second, more severe form of the disease, cysticercosis, is caused by the ingestion of the infective larval stages, called cysticerci, through the oral–fecal route. In severe cases, the larval stages can migrate through the blood–brain barrier leading to seizures and a life-threatening condition known as neurocysticercosis. *T. solium* is one of the most important parasites affecting the central nervous system by disruption of the blood–brain barrier [17,18].

The lifecycle of *T. solium* involves humans as the definitive hosts, harboring the adult tapeworm, and pigs as the intermediate host harboring the infective larval stages [16]. In humans, the gravid segments are passed out into the environment via feces, which may then be ingested by pigs. Within the porcine alimentary tract, the eggs rupture, releasing oncospheres that penetrate the gut and, via the portal vein, reach vital organs before settling in various muscle tissues. Within the porcine host tissues, oncospheres develop into cysticerci featuring the scolex or head of the tapeworm. Upon consumption of raw pork containing cysticerci, the lifecycle continues within the human alimentary tract, where the tapeworm matures sexually and produces eggs which are then passed out in feces.

The literature on in vitro cultivation of *T. solium* is sparse. In vitro studies have employed *Taenia crassiceps* (ORF strain cysticerci) as a closely related experimental model species to *T. solium* [19,20]. Standard cell culture involves the cultivation of *T. crassiceps* cysticerci in RPMI medium [21]. In order to elucidate the mechanisms of *T*. *solium* oncosphere attachment to host intestinal tissues, Verastegui et al. [22] established in vitro models using three substrates: porcine mucosal scrapings with mucin; porcine mucosal explants; and monolayers of CHO-K1 (Chinese hamster ovary) cells. All substrates were incubated with activated oncospheres for 1.5 h, followed by washing and counting of oncospheres still adhering to the monolayers; among the various substrates tested, the best results were obtained in the case of CHO-K1 monolayers. In addition, the number of attached oncospheres was significantly higher for viable monolayers in comparison to fixed monolayers. Although the CHO monolayer model provides a valuable tool for studying parasite attachment, due to the lack of a 3D structure, as seen in the host, this system is unable to simulate the native host environment accurately.

Two recent publications reported the co-culture of activated oncospheres to obtain postoncospheral forms at various stages of development [23,24]. However, *T. solium* oncospheres were viable only for a maximum of 60 days in HCT-8 cells. Both studies carried out infectivity assays in rats where the postoncospheral forms were shown to develop into mature cysticerci. Another recent publication reported the identification and culture of proliferative cells in abnormal larvae of *T. solium* with implications for a better understanding of the racemose form of cysticersosis affecting the brain and spinal cord in humans [25].

##### *Echinococcus* *granulosus*

*Echinococcus (E.) granulosus*, also known as dog tapeworm, has a worldwide distribution and requires a definitive host and an intermediate host to complete its lifecycle [26]. While canines serve as the definitive host, sheep, and other livestock are the intermediate hosts. Infection in humans is known to occur via food and soil contaminated by stool from infected dogs, as well as direct contact with contaminated dog fur. Infections caused by *E. granulosus*, known as cystic echinococcosis (CE), can remain dormant for many years, with hydatid cysts growing to a certain size before rupturing, eventually causing life-threatening conditions, such as severe hepatic and pulmonary distress, including vascular instability often accompanied by massive cytokine release (cytokine storm) and anaphylactic shock [27,28,29]. Genetic predisposition seems to dictate why some patients are prone to displaying a severe and life-threatening autoimmune response in cases of rupture of liver cysts [30]. In such cases, a functional imbalance in cytokine secretion involving multiple genes appears to play a role in the adverse immune response of the host [31]. The rupture of hydatid cysts releases numerous infective larval stages known as protoscoleces.

Within the duodenum of the intermediate host, the embryos hatch, penetrate through the intestinal wall, and enter the liver and lung tissues via the bloodstream. Upon settling in various organs, the embryo forms hydatid cysts that contain numerous brood capsules and scolices. Fertile hydatids containing fully developed scolices represent the infective stages for the canine host, thereby propagating the lifecycle [26]. 

Strobilization in tapeworms involves transverse fission of the segments, known as strobila, leading to new progeny. The first report outlining optimal conditions for in vitro development of protoscoleces in a strobilar direction, as opposed to a cystic direction *E. granulosus*, was published in 1967 [32]. Since then, several studies have been published on in vitro culturing of *E. granulosus*, especially in the early 1990s. One study using infective protoscoleces obtained from sheep was able to show the development of hydatid cysts after 90 days in a medium supplemented with glucose, glutamine, and fetal calf serum at 37 °C; at this time, echinococcus cysts developed in vitro showed a five-fold increase in size [33]. Various developmental stages of *E. granulosus* were obtained from this study, and the cysts were then used for infectivity studies in mice, where 15 cysts surgically implanted in each mouse gave rise to an average of ten cysts at 210 days following infection. This study showed that in vivo infection with cysts developed in vitro was a valuable approach when compared to infection using protoscoleces since the cysts owing to their laminar layer were better adapted to evade the host immune system.

Prolonged culture of the strobilar stage of *E. granulosus* was maintained in a study involving parasites obtained from two distinct animal sources [34]. This study showed marked differences in the success of long-term culturing of these parasitic worms: protoscoleces obtained from sheep developed into mature, segmented worms between 81–114 days of culture, while those sourced from donkeys were unable to mature beyond the initial stages after 67 days of culture. A major obstacle toward the establishment of the complete *E. granulosus* life cycle was overcome with the first report of successful in vitro insemination showing an egg containing a striated embryophore representing the initial stage toward the formation of a fertilized egg in the uterus cavity of a worm maintained in culture for 82 days [35].

An exploratory study examined the differences in the expression of five genes, chosen due to their role in the development and sexual maturation of *E. granulosus* in ex vivo harvested and in vitro grown parasites [36]. The results of the qPCR analysis of this small subset of genes revealed large differences in mRNA transcription levels between the ex vivo and in vitro parasite test groups. As concluded by the authors, these differences could reflect the disparate environmental influences on the ex vivo and in vitro samples that they were exposed to. It would be worthwhile to investigate the fitness, infectivity rate, and life cycle of the acquired in vitro and ex vivo stages on host animal models, as reported previously [37].

##### *Echinococcus* *multilocularis*

Unlike *Echinococcus granulosus*, *Echinococcus (E.) multilocularis* or fox tapeworm produces small, cancer-like cysts or locules in multiple organs of its host, causing a systemic disease named alveolar echinococcosis [38]. Without medical intervention, alveolar echinococcosis is ultimately fatal [39]. The only option for treatment is surgical resection of cysts or chemotherapy involving a combination of mebendazole and albendazole in cases of inoperable cysts. Adult worms are harbored by canids, which act as definitive hosts, while mice and voles are the intermediate hosts harboring the larval stages [16]. Foxes (*Vulpes vulpes*), the primary definitive hosts, become infected by ingesting the cyst-containing organs of infected rodents, which are intermediate hosts. Humans are accidental intermediate hosts becoming infected by ingestion of parasite eggs. Within the human intestine, oncospheres are released, followed by the development of cysts in the liver tissue.

One of the earliest studies published reported on the production of eggs in adult worms of *E. multilocularis*, following partial development in the definitive host for 20–21 days and maturation for 8 days in vitro [40]. Taylor and Morris demonstrated in vitro culture of the metacestode or larval stage of *E. multilocularis* [41]. Another simple, easy-to-handle model involved placing blocks of tissue from experimentally infected mice into a suitable medium supplemented with fetal calf serum, with the appearance of vesiculated structures on the tissue blocks, which after about two weeks developed into metacestodes, an important developmental stage, that was capable of proliferation and differentiation [42]. In vitro culture of *E. multilocularis* under axenic conditions was performed using reduced medium conditions, and metacestode vesicles were generated by co-culturing with rat Reuber hepatoma cells for up to three weeks [43]. This method, first reported by Spiliotis et al., has been used recently for culturing *E. multilocularis* [44,45]. 

A more recent publication described the development of a novel in vitro model for cultivation of *E. multilocularis* protoscoleces in a modified RPMI-1640-based medium, with 25% (*v*/*v*) fetal bovine serum [46]. Following long-term culture (100 days), the protoscoleces developed into larval cysts. This study represents an easy-to-implement cell-free in vitro culture method for obtaining protoscoleces using RPMI-1640 medium without the need for host feeder cells.

Newer research describing a 3D hepatocyte culture system for larval development in *E. multilocularis* is a significant step in studying parasite development, as well as host–parasite interactions, as it enables long-term preservation of hepatocytes when compared to monolayers [47]. The 3D co-culture model involved a 5:1 mix of hepatocytes and mesenchymal stem cells (MSCs), respectively, on a collagen scaffold in the presence of 2D and 3D hepatocyte co-cultures and MSCs (1 × 10^5^ cells), and maintained in DMEM media supplemented with 10% FBS (*v*/*v*), along with several supplements to aid the development of metacestodes. At 2-day intervals, half the medium of the co-culture system was replaced, with the entire system being constantly shaken in a shaker. The 3D co-cultures were compared to 2D cultures and controls, as seen in Figure 1.

### 3.2. Trematoda

Parasitic flukes of the class trematoda uniquely have mollusks as the most common intermediate hosts in their life cycles. With the exception of Schistosomes, all trematodes are monoecious with alternating sexual and asexual cycles.

#### 3.2.1. *Clonorchis sinensis*

Clonorchiasis, caused by the human liver fluke *Clonorchi (C.) sinensis*, is endemic to Southeast Asia. Chronic infections lead to cholangiocarcinogenesis, a fatal liver condition involving malignant tumors of the epithelium of the biliary tract [48]. The lifecycle of *C. sinensis* involves three hosts, one definitive host (man), the first intermediate host (snail), and the second intermediate host (fish) [16]. Human transmission occurs via ingestion of encysted metacercariae, which are the infective stages present in contaminated fish [49]. Excystation of the metacercariae occurs in the host duodenum, and within a few weeks, maturation occurs, releasing adult worms [50].

An early publication by Sun et al. on the in vitro maintenance of adult forms of *C. sinensis* obtained the longest survival times of these parasites using medium 199 (Gibco^TM^, Waltham, MA, USA), with the addition of 5% rabbit serum [51]. Uddin et al. studied egg production and viability under various medium conditions by adult worms maintained in long-term culture [52]. The same research group also examined in vitro maintenance of adult worms of *C. sinensis* in various media, including inorganic solutions, nutrient media, and supplemented 1× Locke’s solution. They concluded that among the inorganic solutions tested, Locke’s solution demonstrated the best results with the survival of the parasites for up to 57 days. Among the various nutrient media tested, RPMI-1640 was the most ideal, with parasite survival ranging up to 114 days in the culture [53].

A recent publication reported the successful development of a 3D in vitro culture model (Figure 2) simulating the microenvironment in a bile duct on an extracellular matrix comprising cholangiocarcinoma cells cultured on a type I collagen hydrogel using a microfluidic device [54]. Another in vitro study employed a 3D cholangiocyte spheroid culture model that contributed to the understanding of pathological mechanisms by simulating prolonged and repetitive clonorchiasis infections [55]. H69 cells maintained in Dulbecco’s Modified Eagle Medium (DMEM) between the 25th and 30th passage were seeded onto InCyto Co. SpheroFilm™ microwells (Chonan, Korea), placed at the bottom of 60-mm culture dish (Falcon, Corning, NY, USA), and incubated with complete culture medium at 37 °C in 5% CO_2_ for 24 h. On day 5 of culture, the fully formed spheroids in each microwell were incubated with serum-/EGF-free culture (conditional) medium for 24 h, followed by ESP treatment, carried out according to the protocol described in the publication of Won et al. [54] above resulting in the harvest of approx. 50 spheroids on days 5 and 10. 

#### 3.2.2. *Fasciola* spp.

*Fasciola* (*F.*) *hepatica* and *F. gigantica* are the two trematode species causing zoonotic liver disease, fascioliasis (fascioliosis). According to the WHO, human infections were rare in the past, but an increasing number of cases are now being reported worldwide for *F. hepatica* in Europe and both *F. hepatica* and *F. gigantica* in Asia and Africa, respectively [56]. Of special research interest are the hybrid forms showing intermediate characteristics of both species, which have been found in endemic parts of Asia and Africa. CDC implicates the import of contaminated produce toward disease transmission in non-endemic regions, especially in Europe. Due to the increased transmission of fascioliasis in recent decades, it is considered one of the more recent model examples of re-emerging diseases [57]. Resistance to the drug of choice for the treatment, triclabendazole, has increasingly been reported; however, the exact mechanism of drug resistance is poorly understood [58]. The lifecycle of *F. hepatica* involves humans and cattle as definitive hosts, sheep as reservoir hosts, and snails of the genus *Lymnaea* as intermediate hosts. Human infection occurs by ingestion of metacercariae via aquatic vegetables, such as watercress from infested waters [16]. 

Hanna and Jura reported the in vitro culture of *F. gigantica* on tissue culture feeder layers, followed by infectivity studies in mice [59]. Although no further growth occurred in the in vitro system, the juvenile parasites were maintained for up to 6 weeks. A more recent publication employed a modified horizontal diffusion system using improvised Eppendorf tubes (Figure 3) for studying the migration patterns of newly excysted juveniles of *F. hepatica* [60]. The horizontal diffusion chamber consisted of an apical compartment made using an inverted Eppendorf tube, the basolateral compartment containing vacutainer tubes; jejunum tissue was mounted via an aperture on the tube lid; juvenile parasites were introduced through an aperture at the top of the construction, and the entire system was maintained under the temperature-controlled settings. A novel in vitro model exploring the interaction between newly excysted *F. hepatica* juveniles and the host intestinal epithelium on a molecular level has been established recently [61]. Excystation of *F. hepatica* metacercariae to obtain newly excysted juveniles was carried out, and optimal culture conditions for the expansion of mouse primary small intestinal epithelial cells (MPSIEC) were optimized.

#### 3.2.3. *Schistosoma* spp.

Schistosomiasis, also known as Bilharzia, is caused by parasitic blood flukes of the genus *Schistosoma* and is associated with tropical and sub-tropical countries, especially in areas with limited access to safe drinking water and sanitation facilities. The primary infective species include *Schistosoma (S.) haematobium*, *S. japonicum*, and *S. mansoni*. Snails act as the intermediate hosts for these dioecious (separate sexes) trematodes, entailing a complex life cycle requiring several weeks for completion. Human infection occurs by penetration of the skin by the cercariae in infested water bodies [16].

Although livestock and other mammals may serve as reservoirs for schistosomiasis, it has long been thought that *Schistosoma* species are strictly host-specific [62]. However, newer studies investigating the zoonotic potential of schistosomes suggest the emergence of hybrid species capable of switching between animal and human hosts [63]. The lifecycle involves two water-borne stages, miracidia, infective to intermediate hosts (snails), and cercaria, shed by snails and infective to mammalian hosts. WHO aims to eradicate schistosomiasis globally. The indicated time frame has now been shifted to the year 2030 [64,65]. 

There has been a fair amount of research in the field of in vitro cultivation of *Schistosoma* spp. Yoshino et al. [66] emphasized that in vitro culture of schistosomes is especially valuable in the study of parasite development by providing a platform for the manipulation of these parasites at genetic and molecular levels. Ye et al. tested the effect of co-culturing *S. japonicum* with various vertebrate host cells and concluded that human hepatic venous endothelial cells (ED25) were most favorable for the cultivation of the lung-stage of these parasites [67]. Another publication described a detailed step-by-step procedure for the isolation of miracidia from infected mice livers, followed by the in vitro transformation of the miracidia into primary sporocysts [68]. Subsequent steps in the lifecycle were carried out in an in vitro setting, where sporocysts, via asexual reproduction, were induced to form secondary sporocysts, which further developed into cercariae capable of infecting humans. 

Frahm et al. published a robust in vitro cell-free culture method for generation and long-term maintenance of *S. mansoni* larval developmental stages, including skin, lung, as well as liver stages. The addition of human serum to the system enabled further development into advanced liver-stage worms phenotypically similar to ex vivo harvested adult worms. Therefore, this system provides a reliable alternative to animal experiments in the parasite development [69].

### 3.3. Nematoda

The class nematode consists of roundworms, which are ubiquitous in nature, with some members known to cause diseases in humans.

#### 3.3.1. *Ascaris* spp.

Ascariasis caused by *Ascaris (A.) lumbricoides* is one of the most common parasitic intestinal diseases, especially prevalent in young children [70]. The mode of transmission is through direct contact with eggs in contaminated soil or through produce from fields where manure is used as fertilizer. Humans are the definitive hosts for *A. lumbricoides*, and propagation of the species occurs via the transfer of parasite eggs between individuals [16].

To date, only a few publications on the cultivation of *A. lumbricoides* in vitro exist. Many studies have reported using *A. suum* [71], roundworms known to affect pigs, as a substitute organism for *A. lumbricoides*. In the current literature, there is a dearth of information on the complete lifecycle of *A. lumbricoides*, including early embryonic developmental stages, as well as behavior outside of its host. One such study aimed to fill the knowledge gap in the early morphological development of *Ascaris* spp. and embryo viability by microscopic evaluation in an in vitro setting [72]. Twelve stages of development were observed, including the first-ever documentation of pre-larval stages one and two. However, a major drawback of this study was the lack of reliability of the microscopic evaluations for accurate identification of the late-morula and blastula stages due to more time and higher magnification required. Due to the accurate identification of viable early embryonic stages, the present study was successful in lowering the time required for viability testing in comparison to the standard Tulane method [73].

#### 3.3.2. *Trichinella* spp.

Trichinellosis or trichinosis is a zoonotic disease caused by the consumption of raw or undercooked meat products from animals infected with nematodes of the genus *Trichinella*. Trichinellosis poses a serious public health risk, and therefore, countries with developed public health systems have a rigorous testing system in place for pork and pork products from domestic pigs. However, wild boars, as well as other omnivores and carnivores, can carry these parasitic worms, and such game meat should be tested for *Trichinella* parasites. Recent guidelines recommend proper handling and thorough heating of game meat to at least 72 °C core temperature for a minimum of 2 min in order to effectively kill the parasites and make meat safe for consumption [74].

Pigs and wild boars are reservoirs of infection for *Trichinella* in humans. Although the entire lifecycle is completed within the primary animal host, a second host, for example, a human, is required for the continued survival of the parasites. After ingestion of infected meat by humans, the larvae are released within the intestine and grow into adult worms. Following sexual maturation and fertilization, the female worm burrows into the intestinal mucosa and releases the larvae, which then enter into the striated muscle tissue and undergo an encystment [16]. 

The earliest studies describing the cultivation of *Trichinella* worms were in the 1960s. Since then, several attempts to obtain various developmental stages of these parasitic nematodes have been made mainly using mice as animal models. However, an operable in vitro model has been missing. A recent publication described for the first time four molting stages of the parasites using mice intestinal epithelial cell (IEC) monolayers [75]. At 12 h after invasion of the monolayers, the larvae began to molt, and at 24 h, complete cuticle formation was observed. The migration and successful ecdysis of the larvae in the IEC monolayers was observed microscopically.

#### 3.3.3. *Onchocerca volvulus*

Onchocerciasis or river blindness is transmitted by the bites of infected female blackflies (*Simulium ornatum*), which breed on the surfaces of flowing water bodies and causes severe eye and skin complaints, as the common disease name suggests. It is the second leading cause of permanent loss of vision caused by an infective agent [76]. Humans are the only definitive hosts for *Onchocerca* (*O.*) *volvulus*. Microfilariae are transmitted from person to person via the insect vectors (e.g., mosquito bites) [16].

Schiller et al. first developed a system for the in vitro cultivation of *O. volvulus* microfilariae in 1979 [77]. Due to the need for live worms as starting material for in vitro cultivation of these filarial worms, the feasibility of cryopreservation of cutaneous tissues (skin snips and nodules) from patients was tested. Cryopreservation proved to be a viable approach as the parasites obtained from frozen tissues showed no differences in terms of viability and ability to develop further when introduced into an invertebrate or vertebrate host when compared to microfilariae obtained from fresh skin samples.

Voronin et al. [78] published an in vitro cell monolayer platform supporting the development of the infective third stage (L3) to L5 forms and showed that the L5 parasites were capable of maturing into young adults. Cryopreserved L3 larvae were thawed and cultured in 96-well plates containing human Peripheral Mononuclear Blood Cells (PMBC) to obtain motile L4 forms after 6 days of culture. The L4 stages were further cultured using several different monolayer cell lines in 96-well plates for 70 days, with the best growth results obtained in Human Umbilical Vein Endothelial Cells (HUVEC) and Human Dermal Fibroblasts (HDF). In order to solve the issue of entanglement of the L4 larvae (30–40% yield) with the monolayer in the well plates, the researchers used a novel approach of placing a transwell with a polyester membrane insert (Sigma) in the wells seeded with either HUVEC or HDF monolayers. In spite of employing transwells, the cultures with HDF cell lines still presented problems with the entanglement of the parasites with the monolayers, whereas better results were obtained with HUVEC cell lines with weekly medium transfers. 

A novel in vitro platform established recently supported the growth and development of *Onchocerca volvulus* for up to 315 days; during this time, developmental processes, including molting from infective L3 stages into adult parasites, were observed [79].

**Table 1 microorganisms-11-01708-t001:** List of helminthic worms and disease details for which in vitro models have been described in the literature.

	Endoparasite	Disease Caused	Transmission Stage	Primary Mode of Transmission	Treatment in Humans	Cases Worldwide	Disease Related Deaths	DALY Score
Cestoda	*Taenia solium*	Cysticercosis, Neurocysticercosis	Embryonated eggs	Raw or undercooked pork	Albendazole, praziquantel	370,710 ^a^	28,114 ^a^	2,788,426 ^a^
	*Echinococcus granulosus*	Cystic echinococcosis (hydatid disease)	Embryonated eggs	Contaminated food and water	Albendazole, mebendazole, surgical removal of cysts	188,079 ^a^	2225 ^a^	183,573 ^a^
	*Echinococcus multilocularis*	Alveolar echinococcosis	Embryonated eggs	Contaminated food and water	Surgery, albendazole	18,451 ^a^	17,118 ^a^	687,823 ^a^
Trematoda	*Fasciola* spp.	Fascioliasis	Metacercariae	Aquatic vegetables from contaminated water bodies	Triclabendazole	10,635 ^a^	0 ^a^	90,041 ^a^
	*Schistosoma* spp.	Schistosomiasis (Bilharzia)	Free-swimming cercariae	Skin contact with infested water	Praziquantel	240,000,000 ^b^	24.072 ^c^	2.543.364 ^c^
	*Clonorchis sinensis*	Clonorchiasis	Metacercariae	Raw or undercooked fish	Praziquantel	31,620 ^a^	5770 ^a^	522,863 ^a^
Nematoda	*Ascaris* spp.	Ascariasis	Embryonated eggs	Contaminated soil, produce	Albendazole, mebendazole	760,000,000 ^d^	6.248 ^c^	1.433.408 ^c^
	*Trichinella* spp.	Trichinellosis/Trichinosis	First-stage (L1) larvae	Raw or undercooked meat	Albendazole, mebendazole	4472 ^a^	4 ^a^	550 ^a^
	*Onchocerca volvulus*	Onchocerciasis (river blindness)	Third-stage (L3) filarial larvae	Bites of infected female blackflies (*Simulium* spp.)	Ivermectin	20,900,000 ^e^	2 ^c^	962.425 ^c^

DALY = Disability-Adjusted Life Year score; ^a^ Togerson et al. (2015) [80]; ^b^ World Health Organization Schistosomiasis (Bilharzia) [65]; ^c^ World Health Organization Global Health Estimates (2018) [81]; ^d^ Betson et al. (2014) [82]; ^e^ World Health Organization Fact sheets Onchocerciasis [83].

## 4. Discussion

Even in the 21st century, mainly due to neglect and re-emergence, parasites continue to remain a major cause of disease, suffering, and fatality in humans. With improvements in sanitation, quality of food, and clean drinking water, as well as access to better overall hygiene practices, the occurrence of parasitic diseases and, hence, the accompanying disease burden can be significantly reduced [84]. The silent epidemic of parasitic diseases and the associated disease burden remains largely hidden as not all helminthic diseases are associated with high mortality rates (YLL); nevertheless, the widespread and crippling disabilities (YLD) associated with these conditions must be fully taken into consideration in order to improve overall global health.

As detailed in Table 1, there are a limited number of anti-parasitic drugs available to treat debilitating parasitic diseases in humans to date. The same standard set of anthelmintic drugs is routinely prescribed against helminths, varying widely in terms of the clinical disease picture presented. Progress has been slow toward the discovery and testing of new drugs, thereby limiting the options for treatment available to clinicians. Successful candidates from veterinary medicine could broaden the available repertoire for human treatment and should be investigated further for their ability as alternative treatment options in humans.

In recent years, new in vitro models in the field of helminth research, especially for *Taenia solium* and *Taenia saginata* [23], have been published. On the other hand, for such parasites as broad fish tapeworms belonging to *Dibothriocephalus* spp. (*syn. Diphyllobothrium* spp.) in vitro culture has been unsuccessful [85]. Diphyllobothriasis, caused by closely related species of the broad fish tapeworm, is a re-emerging parasitic disease outside endemic regions mainly due to increased trade and consumption of raw fish in conjunction with the rising popularity of exotic cuisines [86]. 

Keeping in mind such socio-ecological factors as urbanization, deforestation, encroachment into hitherto untouched areas, population growth, and environmental effects associated with climate change occurring at a rapid pace, the importance of emerging and re-emerging diseases, including often-neglected parasitic diseases cannot be ignored. Therefore, this review includes parasitic diseases that cause the most important NTDs [87], for which in vitro models have been published. Among the parasitic NTDs, Schistosomiasis caused by *Schistosoma* spp. is the second most socio-economically devastating parasitic disease after malaria [88]. A recent paper reported an alarming increase in the number of NTDs in Italy due to various factors, including migration, international tourism, and climate change [89]. There is an urgent need for policy change in order to educate and prepare healthcare professionals for the unmitigated spread of parasitic NTDs beyond their known geographical hotspots.

There is no doubt that in vitro culture supplies valuable insights into the lifecycle and mode of transmission and provides parasite material for drug testing [90]. In addition, recent advancements in 3D culturing methodology, such as hollow fiber technology, 3D tissue printing, or spheroid-based matrices, are promising developments in pursuing and improving in vitro cultivation, including helminths. Nevertheless, as for all in vitro environments mimicking in vivo conditions, the question remains how comparable these in vitro-generated parasitic stages to those occurring within the natural environment of their hosts are [90]. During our assessment of recently established in vitro models in the field of helminthology, it became evident that at the current stage of in vitro culture technology, challenges in establishing an in vitro system remain high, especially with increasing complexity of the go-to organisms. This is certainly the case for helminth in vitro systems, and to this day, human pathogenic helminths without an in vitro model outnumber those with one. Development of helminth in vitro models often is time and resource-consuming, and funding opportunities for NTDs caused by helminths are limited, further slowing development progress.

All of the presented models were successful in maintaining at least one developmental stage of the parasite. To this day, the development of an ex vivo model achieving complete maintenance of a parasite and all of its development stages remains elusive, and even with the current fast-paced technology progress, it remains questionable if such a system may ever be established. Therefore, efforts should remain focused on the infectious stages of these parasites, and as technological opportunities advance, further efforts that also address the omission of animal models in a replace, reduce, and refine (3R) manner can be looked forward to as proof of concept studies. The present review is not intended to be a systematic listing of all the publications on the very broad topic of the in vitro models of helminths relevant to human health but intended to be a go-to collection for recent advances of successful methodology in this field. To our knowledge, this review is the first of its kind to provide a comprehensive overview of the state of recent research on available in vitro models for severe disease-causing parasitic worms from the perspective of each parasite.

## 5. Conclusions

To our knowledge, this is the first review to offer a qualitative summary of existing in vitro models for helminths with significant global disease burden. Publications highlighting significant advancements and turning points for each of these parasites were included. In vitro models provide the following advantages: the ready availability of parasitic lifecycle stages for further research; platforms for testing chemotherapeutic agents; and ethical alternatives to in vivo experiments. Further research and interest from the scientific community and governmental organizations are needed to advance the current state of helminth research with the goal of facilitating discussions about preventing and treating crippling parasitic diseases affecting the global population.

## Figures and Tables

**Figure 1 microorganisms-11-01708-f001:**
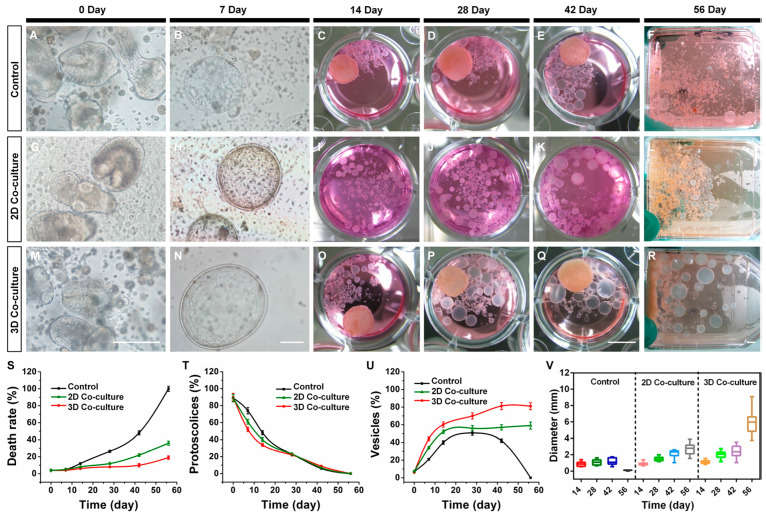
A composite image showing various developmental time points in the 3D hepatocyte culture system for *E. multilocularis* in comparison to 2D culture and control. The figure shows the following insets: (**A**–**F**) development of protoscoleces cultured in the control group; (**G**–**L**) 2D co-cultured group; and (**M**–**R**) 3D co-cultured group. Insets (**A**,**G**,**M**) show light microscopy images of invaginated or evaginated protoscoleces on day 0 of culture; scale bar: 100 μm. Insets (**B**,**H**,**N**) show de-differentiation and microvesicle formation for *E*. *multilocularis* protoscoleces after 7 days of culture, scale bar: 100 μm. Insets (**C**–**F**), (**I**–**L**), and (**O**–**R**) show light microscopy images of developed vesicles after culturing 14, 28, 42, and 56 days of culture, respectively; scale bar: 5 mm. In addition, the following insets are depicted: death rate of protoscoleces (**S**); percentage of protoscoleces (**T**); percentage of vesicles (**U**); diameter of the vesicles (**V**) in the control group, 2D co-cultured group, and the 3D co-cultured group in the duration of culture. Image reused from the open-access publication by Li et al. (2018) [47].

**Figure 2 microorganisms-11-01708-f002:**
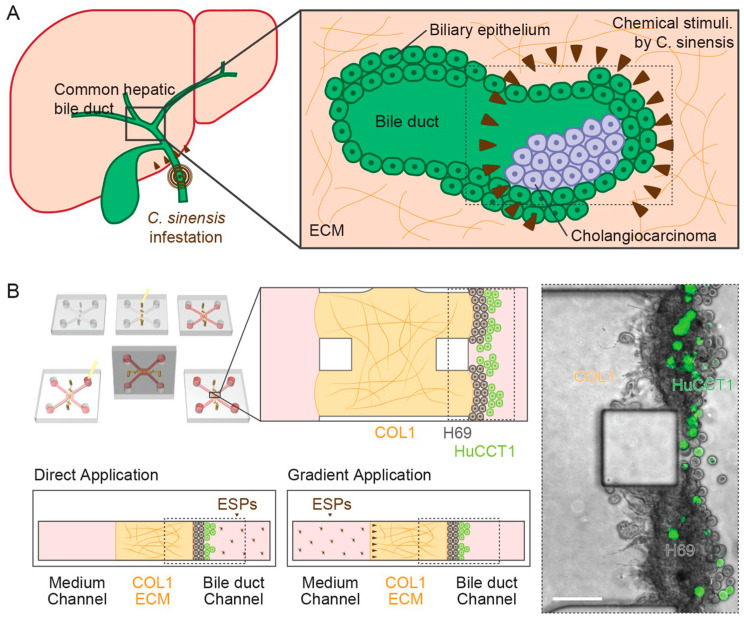
A schematic representation of the 3D co-culture of *C. sinensis*, with biliary ducts using COL1 hydrogel scaffolds. Inset (**A**) (left) hilar cholangiocarcinoma, a form of hepatic bile duct cancer and human liver infected by *C. sinensis*, is on the left. Inset (**A**) (right) formation of a tumor gland in the bile duct and brown triangles depicting chemical stimulation by excretory–secretory products (ESPs) of *C. sinensis*. Inset (**B**) depicts a COL1 hydrogel scaffold region for cell culture. Bottom-right shows a Phase-contrast image with green fluorescence protein-expressing HuCCT1 CCA cells in a tumor microenvironment (boxes with dotted lines) consisting of H69 normal cholangiocytes showing the application of ESPs. Image reused from the open-access publication Won et al. (2019) [54].

**Figure 3 microorganisms-11-01708-f003:**
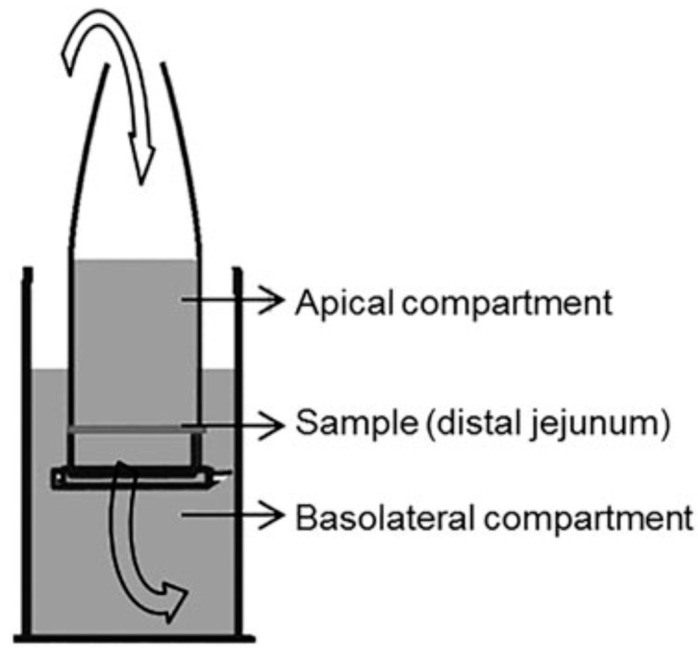
Diagrammatic representation of the modified horizontal diffusion system employing an inverted Eppendorf tube migration studies of newly excysted juveniles of *F. hepatica*. The system consisted of an apical and a basolateral compartment. For the apical compartment, an eppendorf tube was used. The basolateral compartment was made up of half-filled 9 mL vacutainer tubes inside a temperature-controlled system. Newly-excysted juveniles (NEJ) *F. hepatica* were introduced via an aperture drilled into the tip of the apical chamber. Another aperture was drilled into the lid of the chamber where the tissue sample was mounted. NEJ traversing through the jejunum were collected from the bottom of the basolateral compartment. Image reused from the publication Garcia-Campos et al. (2016) [60]. Permission to reuse image obtained from copyright holder.

## Data Availability

No new data were created or analyzed in this study. Data sharing is not applicable to this article.

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
