# Peer review of "The Truman Show for Human Helminthic Parasites: A Review of Recent Advances in In Vitro Cultivation Platforms"

_microorganisms, 2023, doi:10.3390/microorganisms11071708_

Round 1
Reviewer 1 Report
This review discusses a critical point in the parasitic diseases (in vitro culture of helminths). There are some minor issues in this review:
1. The authors didn't show the obstacles for prevent or limit the efforts to design suitable cultures for helminths. So, what are these obstacles?
2. I think the authors could cite any work related to veterinary parasites like Haemonchus contortus
3. There is no any data about Ancylostoma doudenale
4. What are the perspectives that help in developing the in vitro culturing of helminths
Author Response
We thank the reviewer for the appraisal of our manuscript and the valuable suggestions provided therein.
We have revised the manuscript according to your valuable suggestions.
A point-by-point response can be found within the submitted letter.
Best regards
Martin Richter

Reviewer 2 Report
General comments:
The subject of the manuscript is important for two aspects: the number of publications is limited and highlights the importance of alternative procedures aligned with the 3Rs (Replacement, Reduction and Refinement) principles for ethical animal research in Parasitology. The manuscript is too long because it contains excessive information with limited contributions to the described objective (see coments above).The manuscript could be shortened with less information about the species described.
Spefic comments:
Introduction section:
Since the purpose of the study is to evaluate helminth models, the text described and the references (lines 34-37; 40-41) refer to the occurrence of parasitosis caused by protozoa, the elimination of this information can make the introduction more objective. Although Caenorhabditis elegans is not a human parasite, it could be highlighted as a conventional model for in vitro model studies in Helminthology.
Material and methods section:
Lines 74-76 - What criteria are used to define possible species, where information on in vitro culture is available? For instance, there are publications on the in vitro cultivation of species of the genus Ancylostoma
Authors must inform the time interval of publication from the first to the last year of analysis.
Was there any exclusion criteria?
Authors should report how many manuscripts were identified according to each search index.
Results section:
Lines 80-83 - This information does not result from the methodology. It should be included in the introduction, instead of the information about protozoa.
Line 85 – According to taxonomy, Cestoda is a class, not a subclass.
Lines 150-151 - There is a repetition of information already presented on page 3 in lines 135-138.
Line 135- Hydatid worm is an inappropriate nomenclature.
Line 192- Echinococcus multilocularis or fox tapeworm
Pg. 6 – line 246 - According to taxonomy, Trematoda is a class, not a subclass.
Pg. 9 – line 346 - WHO guideline on control and elimination of human schistosomiasis (2022) indicated by 2030.
Line 366- According to taxonomy, Nematoda is a class, not a subclass.
Author Response
Thank you for this highly valuable and thorough appraisal of our manuscript. Please find our point-by-point remarks and manuscript adjustments as of your comments in the attached file.

Reviewer 3 Report
This review highlights the development of in vitro models to study different helminth groups. Please do accept my personal comments and suggestions as follows:
I encourage authors to go further in the overall analysis of the advances of in vitro models within each helminth group. I believe discussion section require a more in-depth assessment of these advances, may be raise questions given this is a really interesting topic: Which field (type of helminth group) is more advanced than the other? or for what type of helminth group do we face a total lack of progress? For instance, recent advances for in vitro models study, the so called “organoid technology” have revolutionized the in vitro culture tools for biomedical research by creating three-dimensional (3D) model. This new organoid technology enables to recreate more complex in vitro models. Has this progress been applied to helminths study? Is this the future for in vitro models study in the field of parasitology? (helminthology).
Therefore, I suggest authors to discuss future opportunities and challenges on this field, which is something we always expect from a review article, suggest ways in which further insights could be gained, what are the key points for further in vitro model advances for the different helminths groups.
Minor comments:
Line 90: “cysticerci”. As it is well known, this type of larval stage containing 1 single invaginated scolex is not the only one occurring in taeniids. In fact other taeniids such as Echinococcus or Taenia multiceps develops into hydatid cyst and Coenuros cerebralis respectively, containing high numbers of protoscoleces. Therefore I suggest using another terminology as it may mislead the reader
Line 152: cattle may act as intermediate host but sheep is more important as intermediate host than cattle (from an epidemiological point of view). So I do believe it is always preferable to refer to sheep as the main intermediate host
Line 463: I do not agree with the statement that there are limited number of anti parasitic drugs agaisnt debilitating parasites. In fact, Table 1 does not illustrate the broad spectrum of drugs available for all parasites included in that table. For instance, against Fasciola spp, there are a wide variety of antiparasitic drugs, not only the common benzimidazole type of drugs but others such as Closantel, Clorsulon, Raxoxanide, Nitroxinil, etc; or for instance piperazine is still widely used against Ascaris infections in many countries. In addition, the antiparasitic drugs market is projected to grow with a CAGR of 5.1% during the forecast period 2022 - 2027. Therefore, I suggest authors to reanalize that statement in terms of limitation of antiparasitics drugs
Diphyllobothriasis is only mentioned in discussion and was not previously assessed in the cestoda section analysis.
Author Response
We thank the reviewer for the appraisal of our manuscript and the valuable suggestions provided therein. Please find a detailed point by point response in the attached file. We hope to have answered all the valuable suggestions/requests satisfactory.

Reviewer 4 Report
The proposed manuscript is an important and interesting work on the various in vitro models identified in the literature for highly transmissible helminthic parasites of medical interest.
It is well written, documented, and the scientific relevance is important: I agree with the publication after minor changes :
- In vitro in italics in "Keywords" section please;
- Plathelminths instead of "Platyhelminths" on line 85.
Author Response
We thank the reviewer for the appraisal of our manuscript and the valuable suggestions provided therein.
Following adjustments were made as of your valuable suggestion:
1) In vitro in italics in "Keywords" section please;
A: in vitro is now italicized. Thank you for spotting this.
2) Plathelminths instead of "Platyhelminths" on line 85.
A: Thank you for spotting this. Changes were made according to reviewers' suggestions.
Reviewer 5 Report
This paper offers a comprehensive overview of the available in vitro platforms to be used in research activities studying parasites that severely affect humans. The paper is accompanied by nice tables and figures, being the first of its kind. With these being said, I find the paper to be suitable for publication.
Author Response
The authors thank the reviewer for appraisal of our manuscript and subsequent recommendation for publication is highly appreciated.
Best regards
Martin Richter
Round 2
Reviewer 2 Report
The authors answered the suggestions and doubts of the previous evaluation. I suggest a small change in line 157 - Keep only the name dog tapeworm because hydatid tapeworm is not appropriate.
Reviewer 3 Report
Authors have taken into account suggestions and have modified the manuscript accordingly. The manuscript has improved in terms of quality and accuracy. Therefore, I suggest this work to be accepted for publication.